# Weighting of risk factors for low birth weight: a linked routine data cohort study in Wales, UK

Amrita Bandyopadhyay [1], Hope Jones [1], Michael Parker,[1]
Emily Marchant [1], Julie Evans,[2] Charlotte Todd,[2] Muhammad A Rahman,[3]
James Healy,[1,4] Tint Lwin Win,[1] Ben Rowe,[5] Simon Moore [6,7] Angela Jones,[2]
Sinead Brophy [1]

For numbered affiliations see end of article.

**Correspondence to**
Ms Amrita Bandyopadhyay;
A.Bandyopadhyay@swansea.ac.uk

## ABSTRACT

**Objective** Globally, 20 million children are born with a birth weight below 2500 g every year, which is considered as a low birthweight (LBW) baby. This study investigates the contribution of modifiable risk factors in a nationally representative Welsh e-cohort of children and their mothers to inform opportunities to reduce LBW prevalence.

**Design** A longitudinal cohort study based on anonymously linked, routinely collected multiple administrative data sets.

**Participants** The cohort, (N=693 377) comprising of children born between 1 January 1998 and 31 December 2018 in Wales, was selected from the National Community Child Health Database.

**Outcome measures** The risk factors associated with a binary LBW (outcome) variable were investigated with multivariable logistic regression (MLR) and decision tree (DT) models.

**Results** The MLR model showed that non-singleton children had the highest risk of LBW (adjusted OR 21.74 (95% CI 21.09 to 22.40)), followed by pregnancy interval less than 1 year (2.92 (95% CI 2.70 to 3.15)), maternal physical and mental health conditions including diabetes (2.03 (1.81 to 2.28)), anaemia (1.26 (95% CI 1.16 to 1.36)), depression (1.58 (95% CI 1.43 to 1.75)), serious mental illness (1.46 (95% CI 1.04 to 2.05)), anxiety (1.22 (95% CI 1.08 to 1.38)) and use of antidepressant medication during pregnancy (1.92 (95% CI 1.20 to 3.07)). Additional maternal risk factors include smoking (1.80 (95% CI 1.76 to 1.84)), alcohol-related hospital admission (1.60 (95% CI 1.30 to 1.97)), substance misuse (1.35 (95% CI 1.29 to 1.41)) and evidence of domestic abuse (1.98 (95% CI 1.39 to 2.81)). Living in less deprived area has lower risk of LBW (0.70 (95% CI 0.67 to 0.72)). The most important risk factors from the DT models include maternal factors such as smoking, maternal weight, substance misuse record, maternal age along with deprivation—Welsh Index of Multiple Deprivation score, pregnancy interval and birth order of the child.

**Conclusion** Resources to reduce the prevalence of LBW should focus on improving maternal health, reducing preterm births, increasing awareness of what is a sufficient pregnancy interval, and to provide adequate support for mothers' mental health and well-being.

## STRENGTHS AND LIMITATIONS OF THIS STUDY

⇒ This study has built an e-cohort using data-linkage across multiple routinely collected administrative data sets to investigate the risk factors of low birth weight (LBW) for the population of Wales.
⇒ The study has investigated the modifiable risk factors of LBW in a holistic framework by linking primary and secondary care physical and mental health, socio-demographic and pregnancy-related routine data including police record for a nationally representative sample.
⇒ This study undertook two different statistical approaches (regression analysis and data-driven machine learning algorithm) which is a strength of the study.
⇒ This work was unable to include any important risk factors which were not recorded in the healthcare system or any conditions which were undiagnosed hence that did not result in the system.

## INTRODUCTION

The WHO defines low birth weight (LBW) as infants weighing less than 2500 g (5.5 pounds) irrespective of gestational age.[1 2] Latest figures show that each year around 53 000 live births (6.9%) are identified as LBW in the UK.[3] LBW is the result of intrauterine growth restriction (less than 10th centile of weight for sex and gestational age), prematurity (gestational age less than 37 weeks) or a combination of both.[4] LBW can impair the baby's cognitive development and lead to developmental disabilities and poor academic achievement.[5] Furthermore, LBW significantly increases the risk of perinatal and neonatal mortality and longstanding morbidity in early and later life.[6] While there has been a reduction in mortality among preterm infants in the last two decades, the incidence of preterm birth has increased in many developed countries.[6–8] The increase is also associated with preterm delivery of multiple pregnancies,

with medically indicated preterm birth 10 times more likely in multiple pregnancies than singleton births.[9] To address the global burden of LBW, the 65th World Health Assembly Resolution 65.6 endorsed a comprehensive implementation plan to achieve a 30% reduction in LBW by 2025.[1] A study conducted on the birth data from 148 countries of 195 United Nations' member states indicated that there had been a 2.9% reduction in the LBW prevalence in 2015, compared with 2000 worldwide. However, there has not been any change in the LBW prevalence in high-income regions (including Europe) and the progress is slower than required to meet the WHO LBW target by 2025.[10]

Existing research has found factors linked with mothers, such as age, high deprivation and low academic qualification, are associated with increased odds of LBW.[11 12] Modifiable risk factors for LBW include interpregnancy interval,[13] maternal physical[14–17] and mental health[18 19] and environmental exposures during pregnancy.[20] Studies have also shown numerous health behaviours such as smoking,[21 22] alcohol intake (in which there is a dose-response relationship with LBW)[23] and/ or illicit drug use[24] during pregnancy are modifiable risk factors of LBW. Indirect (negative maternal behaviours, inadequate nutrition or prenatal care and increased stress) or direct (physical assault, sexual trauma) experience of intimate partner abuse during pregnancy can lead to adverse infant outcomes including LBW.[25 26]

It is important to gain an understanding of these risk factors, particularly modifiable risk factors, so that resources and interventions can be scheduled effectively. Moreover, the wide range of risk factors cannot be addressed in isolation. Most of the risk factors that are strongly independently associated with LBW are correlated. This study aimed to understand the contributions of risk factors to the burden of LBW for the population of Wales, using traditional statistical methods and supervised machine learning models.

## METHOD
### Participants and linkage
The linked data cohort (N=693 377) comprised of children born in Wales between 1 January 1998 and 31 December 2018. The study population was identified in the National Community Child Health Database (NCCHD), which is a local Child Health System database held by the National Health Service. The participants were linked to the Wales-wide administrative register, the Wales Demographic Service (WDS) dataset. Linkage was undertaken using an anonymised encrypted linkage key, the anonymised linking field, in the Secure Anonymised Information Linkage (SAIL) Databank.[27] WDS provided the anonymised residential linking fields, which is an encrypted residential address and its corresponding lower super output area (LSOA, small geographical areas with a population of approximately 1500) when the child was born. LSOA was linked with the Welsh Index of Multiple

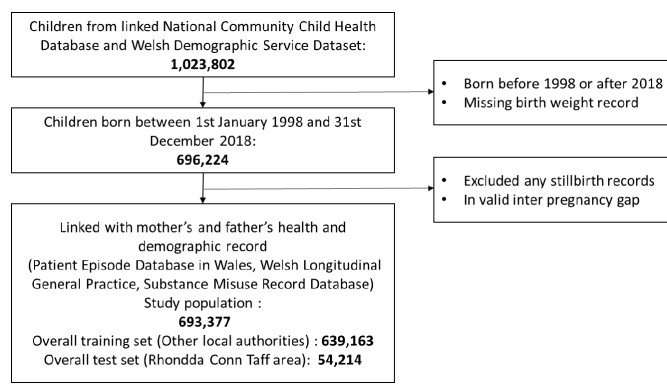

**Figure 1** Participants flow diagram.

Deprivation (WIMD) 2014, which is a measure of relative deprivation. The participants flow diagram is displayed in figure 1.

### Explanatory variables
A literature review was conducted at the beginning of the study to identify the explanatory variables associated with LBW. A study by Johnson *et al* was identified[3] and this provided the framework on which the current study was developed. The literature review selected:
1. Any published systematic reviews since 2013 which focused on risk factors identified in Johnson *et al*.
2. Any published systematic reviews since 2010 for all additional risk factors not identified in Johnson *et al*.

This study therefore considered a wide range of explanatory and confounding variables that have a plausible causal link to LBW and are potentially modifiable at a population level. The literature review to select the explanatory variables has been described in a online supplemental document Supplementary document. In the current study, modifiable risk factors identified from the literature have been derived from routinely collected electronic datasets to build a Welsh e-cohort of the children. The maternal variables related to a childbirth (maternal age, gestational age, child's birth weight, gender and birth order of the child) were obtained from NCCHD and maternal indicator database (MID). The variables for maternal physical (such as diabetes, anaemia, intake of vitamin D and folic acid supplement through prescription) and mental (depression, antidepressant medication, anxiety, serious mental illness such as bipolar disorder, schizophrenia) health during pregnancy were obtained from primary care Welsh Longitudinal General Practice (WLGP) and hospital admissions dataset known as the Patient Episode Database in Wales (PEDW). The record of physical assault linked with mothers during pregnancy was obtained from PEDW. The substance misuse database provided the information on individuals receiving treatment for alcohol and other substance misuse in Wales. Mothers' who were presenting in this database during pregnancy were considered in the study. Area type (urban/rural) and local authority (LA) under which they lived during the pregnancy and their overall and physical environment quantified in the WIMD were

included in this study. A cleaned and harmonised variable of maternal smoking during pregnancy was created based on the data obtained from NCCHD, MID and WLGP datasets. The other derived maternal variables include multiple birth flag (to distinguish between singleton and non-singleton), pregnancy interval and maternal weight. The description of the explanatory variables and their sources have been described in online supplemental table 1.

A subset of the study population (participants from Rhondda, Cynon, Taf, born between June 2016 and 2018) was linked with the Public Protection Notification (PPN) dataset to investigate the impact of the PPN during pregnancy along with other existing risk factors on the risk of LBW.[28] PPN is an information sharing system, completed by police officers that compiles incidents of domestic abuse, stalking or harassment. The current study received PPN data from South Wales Police for residents of South Wales LA Rhondda, Cynon, Taf.

### Outcome variable
A binary variable was created using the birth weight variable obtained from NCCHD.
- ▶ LBW=birth weight <2500.
- ▶ Not LBW=birth weight ≥2500.

### Statistical analysis
It is known that gestational age is highly correlated with LBW. However, as the gestational age is only obtained at the point of birth, making it a non-modifiable risk factor, this study has not considered it as a predictor variable. The models were stratified by the multiple birth as this is one of the main predictors of LBW. The missing records in the birthweight variable were removed from the analysis. Since there was around 15% missing data in the maternal weight variable, the variable was imputed by the simple random imputation method.[29] The missing data in the other explanatory variables (less than 10%) were recorded as 'Unknown'. The birth record for stillbirth and pregnancy interval of less than 22 weeks (as that is the minimum duration for a considerable gestation period) were also not considered for the statistical analysis. Data preparation including data linkage and data cleaning for this analysis was done on SAIL DB2 SQL platform. All statistical analyses were performed in R V.4.0.3.

The statistical analysis of the current study was carried out using two statistical approaches: (a) building a holistic regression model to investigate the association between the risk factors and LBW and (b) building a predictive model using a supervised classification method. Both methods were capable of handling binary outcome variables. The models that were developed by the above-mentioned methods were built independently, however they both were informed by the same dataset. This enabled us to evaluate and validate the findings of the models and helped to gain insight on the generalisability of the findings.

### Logistic regression
A multivariable logistic regression (MLR) model was developed to identify the most important risk factors associated with LBW. The MLR model was built on the overall study population (whole Wales dataset) to examine the associations between all the explanatory and outcome variables. The holistic model considering all the risk factors identified from literature review and selected or derived from routine data includes maternal physical and mental health during pregnancy, maternal smoking, alcohol and other substance misuse record, maternal age, maternal weight, pregnancy interval, living area, LA and deprivation—WIMD score. The MLR model also included the birth order of the child and the multiple birth flag. The birth order highlights the sequential birth position of the child for a mother, and it does not vary among the children who were non-singleton in the same family (please see online supplemental table 1), hence, they were considered as independent variables in the model and their association with the outcome variable was investigated in the MLR model. The importance and significance of the risk factors have been evaluated and presented with their adjusted OR (aOR) and 95% CI.

### Decision tree
A supervised machine learning classifier—decision tree (DT) model was developed to build a risk profile for LBW and test its predictive performance. Classification tree—DT models were constructed using RPART (Recursive Partitioning And Regression Trees) packages in R.[30 31] The algorithm recursively partitions the data into multiple subspaces to obtain the homogeneous final subspace of predictor variables. For DT, the whole Wales data except for Rhondda, Cynon, Taf, was used to train the model and prediction performance was evaluated on a test dataset which consisted of a sample of participants from the LA of Rhondda, Cynon, Taf. This LA was chosen because it had one of the highest rates of LBW in Wales and is an area which would benefit most from an accurate prediction model.

A separate data linkage was undertaken with a subset of the study population which was linked to the mother's domestic abuse record from PPN dataset (the latter was only available for Rhonda, Cynon, Taf). Another adjusted MLR model was developed on this linked data to investigate the risk association for LBW.

### Patient and public involvement
No patient involved.

## RESULTS
The study population consisted of 693 377 children of which 54 214 were from Rhondda, Cynon, Taf, and 639 163 were from other LAs. The children from Rhondda, Cynon, Taf, which was later used as a test set for DT were well representative of the Welsh population (see online supplemental table 2). In the overall study

population, 51.26% were boys, 96.92% were singleton and 90.38% children were born full-term (gestational age between 37 and 42 weeks). 49.85% of the children were born as the first child in the family. Mothers of 0.48% children were admitted to hospital for diabetes and 0.09% had a general practitioner (GP) visit for diabetes, 1.27% had depression, 1.52% with anxiety and 0.02% were on antidepressant medication during pregnancy. There were 1.26% and 21.51% children whose mothers had alcohol-related substance misuse and smoking records during pregnancy, respectively. The average maternal age at birth of child and maternal weight was 28 years and 70.82 kg (after imputation), respectively, and 63.68% of them were living in densely populated urban areas. Overall, 7.1% (8.26% in test set and 7% in other LAs) of children were born as LBW.

## Factors associated with LBW: MLR results

Non-singleton children were at almost 22 times higher risk of LBW than singleton children (aOR—21.74 (95% CI 21.09 to 22.40)). Mothers with diabetes-related GP visits (2.03 (95% CI 1.81 to 2.28)) and hospital admission records of anaemia (1.26 (95% CI 1.16 to 1.36)) during pregnancy were at very high risk of having LBW children. Poor mental health during pregnancy such as severe depression (1.58 (95% CI 1.43 to 1.75)), serious mental illness (1.46 (95% CI 1.04 to 2.05)), severe anxiety (1.22 (95% CI 1.08 to 1.38)) and antidepressant medications (1.92 (95% CI 1.20 to 3.07)) were risk factors for LBW. The other highly significant modifiable risk factors linked with pregnant mothers include maternal smoking (1.80 (95% CI 1.76 to 1.84)), alcohol-related hospital admissions (1.60 (95% CI 1.30 to 1.97)) and any substance misuse (alcohol/other drugs) (1.35 (95% CI 1.29 to 1.41)) during pregnancy. Higher maternal age was also associated with the risk of LBW. Though maternal age less than 19 was significantly associated with the risk of LBW in the univariable model, after adjusting all the other explanatory variables, this did not remain as a risk factor of LBW. The first child born was at higher risk of LBW than subsequent births, The odds of LBW for the second child was 0.59 (95% CI 0.57 to 0.60) compared with the first child. Mothers living in the least deprived and rural areas during pregnancy were at lower risk of having LBW children than others living in more deprived and urban areas. The statistically significant risk factors with their aOR and CI have been visualised and described in figure 2 and online supplemental table 3.

## Finding from the linked PPN data model

A data set of 5854 mothers were obtained from the PPN data linkage. Those who had a PPN call during pregnancy, 18% of them had an LBW child and those who did not have a PPN call, 8.7% of them had an LBW child (see table 1). Mothers with a PPN call during pregnancy had almost two times higher risk of having LBW babies (1.98 (95% CI 1.39 to 2.81)) than mothers without PPN call after adjusting for confounding factors (see online supplemental figure 1).

## Predictive DT model

Since LBW were disproportionately more prevalent in non-singleton children (5.61% singleton vs 53.91% of the non-singleton children were LBW) (online supplemental table 4), two separate predictive models using DTs were developed.

### Singleton children

There were 619458 observations in the training model. The most important risk factors selected by the DT algorithm to develop the final tree were maternal smoking, maternal weight, pregnancy interval, birth order, maternal substance misuse record (any), maternal age, deprivation—WIMD score, maternal substance misuse record (other drug) and maternal substance misuse record (alcohol). Online supplemental figure 2 depicts the final tree with the branches including the final 33 terminal nodes. For example, the model would predict an LBW baby if (a) maternal smoking is positive (eg, mum smokes during pregnancy) and (b) maternal weight less than 60 kg. The number of women in this category who had an LBW child is 73% (see terminal node 4 in online supplemental figure 2) and risk profile was found in 7% of the training model population (eg, 7% of pregnant women were smokers who weighed less than 60 kg during pregnancy).

The test data was built on the 52583 singleton children, which is 7.82% of the total singleton children in this study. The model performance is explained in a confusion matrix with 60.54% accuracy, 60.41% sensitivity, 60.55% specificity, 9.68% positive predictive values and 95.63% negative predictive value (see tables 2,3).

### Non-singleton children

There were 19705 children in the non-singleton training subset. The variables selected to generate the tree by the DT algorithm in the importance order were pregnancy interval, birth order, maternal weight, maternal age, gender, deprivation—WIMD score, maternal smoking, living area, deprivation—WIMD (environment) score and maternal substance misuse record (any). Online supplemental figure 3 depicts the final tree with the branches including the final 29 terminal nodes. For example, the model would predict an LBW baby if (a) this is the first child or pregnancy interval is either above 10 years or less than 1 year and (b) maternal weight less than 60 kg (terminal node 4).

The test set was built on the 1631 non-singleton children, which is 7.64% of the total non-singleton children in this study. The model performance was measured as 58.74% accuracy, 68.71% sensitivity, 41.09% specificity, 67.36% positive predictive values and 42.61% negative predictive value (see tables 2,3).

## DISCUSSION

Among the overall study population in Wales 7.1% was LBW between 1998 and 2018. Global trend of LBW is

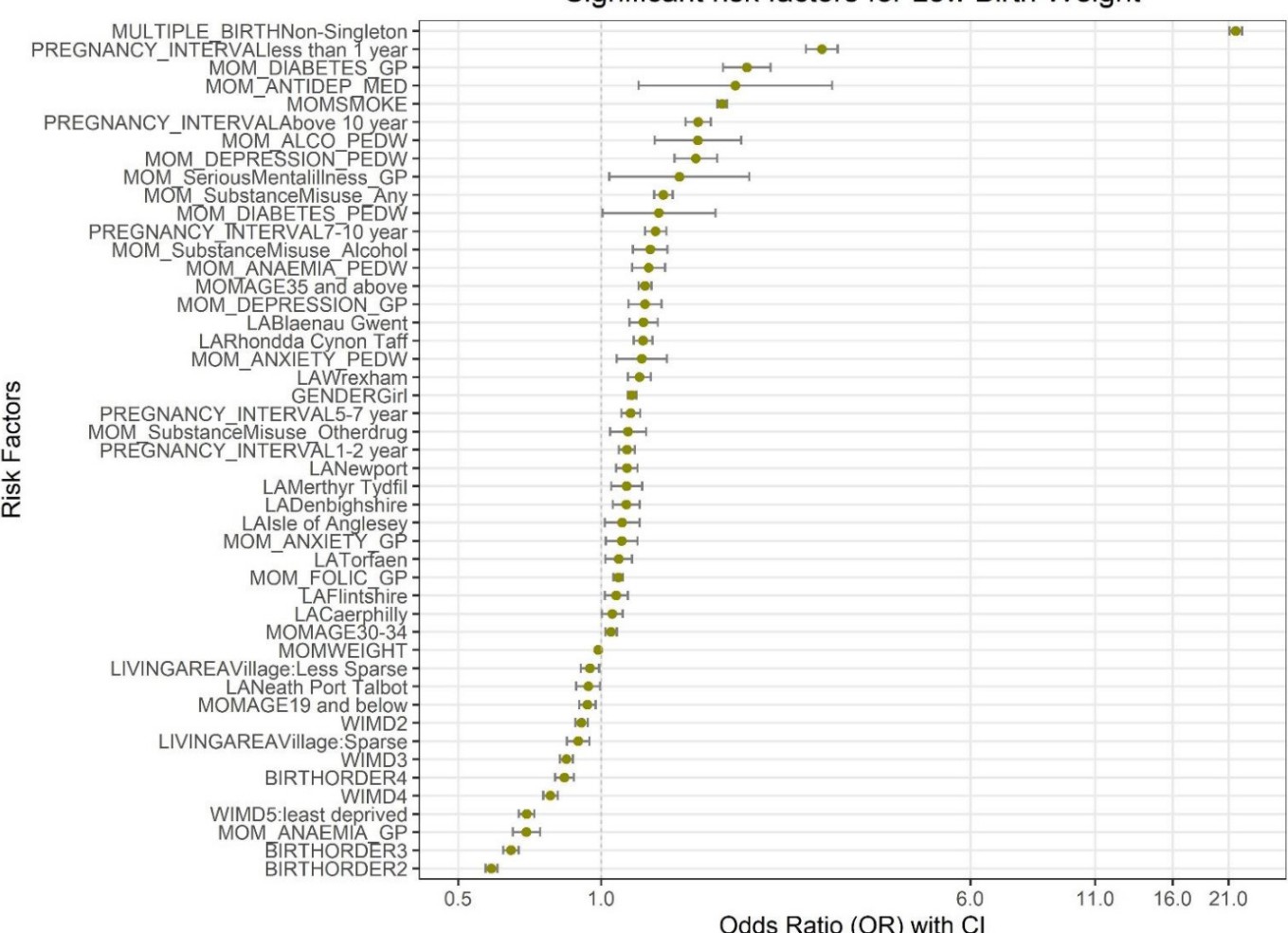

**Figure 2** Significant factors associated with the risk low birth weight among the overall study population. GP, general practitioner; LA, local authority; PEDW, Patient Episode Database in Wales; WIMD, Welsh Index of Multiple Deprivation.

around 7.0% in both 2000 and 2015 for the developed regions (Europe, North America, Australia), which is consistent with our finding.[2] Findings from the Office for National Statistics state a combined English and Welsh rate of LBW of 7.0% in 2016, unchanged from 2011.[32] Our findings show that LBW is strongly associated with non-singleton pregnancy, and maternal health which includes a short pregnancy interval, non-optimal maternal body weight (eg, low, or high weight), maternal smoking, diabetes, anaemia, mental illness and living in a deprived urban area and exposed to domestic abuse during pregnancy.

**Table 1** Distribution of LBW and nLBW children for the subset who were linked with mother's PPN record during pregnancy

| PPN record during pregnancy | n=5854 | |
|---|---|---|
| No | | |
| nLBW | 5074 | 91.3% |
| LBW | 485 | 8.7% |
| Yes | | |
| nLBW | 241 | 82% |
| LBW | 53 | 18% |

LBW, low birth weight; nLBW, not LBW; PPN, public protection notification.

**Table 2** Confusion matrix/two by two table of the decision tree (singleton and non-singleton) models

| Prediction | Reference (singleton) n=52 583 | | Reference (non-singleton) n=1631 | |
|---|---|---|---|---|
| | LBW | nLBW | LBW | nLBW |
| LBW | 2077 (TP) | 19 389 (FP) | 716 (TP) | 347 (FP) |
| nLBW | 1361 (FN) | 29 756 (TN) | 326 (FN) | 242 (TN) |

FN, False Negative; FP, False Positive; LBW, low birth weight; nLBW, not LBW; TN, True Negative; TP, True Positive .

**Table 3** Prediction model performance (n=52 583 singleton, n=1631 non-singleton from test set)

| | Accuracy | Sensitivity | Specificity | Positive predictive value | Negative predictive value |
|---|---|---|---|---|---|
| **DT singleton model** | 60.54% | 60.41% | 60.55% | 09.68% | 95.63% |
| **DT non-singleton model** | 58.74% | 68.71% | 41.09% | 67.36% | 42.61% |

DT, decision tree.

The findings of short and long pregnancy intervals being associated with increased odds of LBW has been reported previously.[13] However, Regan *et al* highlighted that several studies examining long interpregnancy intervals are prone to measurement error because miscarriages and abortions within this time period are difficult to capture. Hence the authors suggest that caution should be exercised when interpreting these findings.[33] Regarding the association of short-pregnancy intervals with increased odds of LBW, studies using matched controlled designs have argued that this association may be weaker than previously thought,[33 34] especially when adjusting for factors such as gestational diabetes, pre-pregnancy obesity, parity and other familial factors.[35] The current study has included diabetes and maternal weight along with pregnancy interval in the analysis. In terms of putting this evidence in context, when considering advice over pregnancy intervals, it will be important to consider all the available evidence including the impact of pregnancy interval on preterm birth and maternal outcomes.[36] Among the modifiable risk factors for LBW identified in this study, smoking during pregnancy is significantly and consistently important. A number of reviews have been carried out in the field of interventions to reduce smoking in pregnancy and this suggest that psychosocial interventions (counselling, feedback and incentives) appear to be effective at supporting women to stop smoking in pregnancy which, in turn, can reduce the proportion of babies born with LBW.[37] However, they argue that the context of the intervention needs to be given consideration and that while evidence exists for potentially effective interventions which could be piloted through delivery of programmes locally, efforts should also be directed at population wide strategies to reduce smoking uptake in young women. This may be especially important given the clear difficulties experienced by pregnant women to give up smoking.[37] With regards to our finding of maternal mental health affecting the risk of LBW, both severe depression and anxiety were associated with an increased odds of LBW in our study.[38]

The study undertook two statistical methods; (a) regression and (b) supervised classification model with the aim that the regression model would identify the risk factors with highest association/OR but not frequently observed factors at the population level for, for example, only 0.09% mothers had diabetes-related GP visit during pregnancy, and they had two times higher risk of having a LBW child (2.03 (95% CI 1.81 to 2.28)). However, the

DT models consider the number of people affected by the risk factor rather than just strength of association, hence capable of identifying the factors at a population level (such as smoking, deprivation score) that can result in higher risk of LBW.

There are similarities between the findings of our DT models and existing literature using machine learning to predict LBW, for example, urban living, higher deprivation and poorer families are at higher risk of LBW.[39] The incidence of LBW in this current work is lower than another research using machine learning to predict LBW, for example, Loreto *et al* has an incidence of 13.45% in work that builds over 60 different machine learning models,[40] Ahmadi *et al* assess logistic regression and random forests in a cohort with LBW rate of 9.5%.[41] The smaller number of active cases in the dataset the more difficult it is to build a prediction model for, particularly without a set of highly associated input variables. In this study, the singleton DT model correctly predicted 60.41% of all the true positive cases. However, the low positive predictive value of 9.68% indicates that the model assigned a false positive 'LBW' classification for 89.32% cases. This model only includes singleton children and since non-singleton pregnancies are highly associated with LBW, removing this variable from the model has lessened its predictive capability. This is evidenced by the significantly improved positive predictive value (67.36%) for the non-singleton model (table 3). Previous machine learning models appear to show better prediction as they included non-singleton, gestational age (which is in terms of temporal association highly associated with LBW but occurs at the same time as the LBW can be measured) and pre-eclampsia in the third trimester. Also, the differences in the proportion of LBW cases, the variables used and the cohort sizes in various other studies alter the ability of the model, hence direct comparison of machine learning models across studies can become difficult.

The strength of this study lies in using a wide spectrum of routinely collected nationally representative administrative data sets of all births in Wales across a large time. This is a very first of its kind study in Wales and adds novelty in the research field of LBW. However, this work can only identify the more severe cases which are recorded in the healthcare system, and undiagnosed cases that did not result in the system will be missed which is a limitation of this work. Since the study was developed on the linked routine data, the limitation of the routine data was encountered in this study, for example, though

the maternal weight variable came from two different sources, data was missing for many participants which was addressed by imputation methods. Also, this study was unable to capture lifestyle factors (diet, physical activity, stress, emotional state) which can be important in determining LBW.[42 43]

The two different models (MLR and DT) used in this study have very similar findings suggesting that factors which are common and so are predictive (using DT methods) such as maternal smoking status and maternal weight could be targeted to address population-level risk of LBW. Factors which have a strong association with LBW (using regression analysis), such as a mother with diabetes or mother on antidepressants as having plausible causal link to LBW, can be addressed to reduce individual risk for that mother/child.

## CONCLUSION

This study suggests that the most important factors to reduce the risk of LBW are to address multiple birth (eg, in assisted reproduction practices), addressing factors associated with preterm births (previous history of preterm birth), addressing maternal health such as reducing smoking, investment in maternal mental health, addressing substance use (alcohol/drugs), treating underlying health conditions (diabetes/anaemia) and promoting planning of pregnancy to give an adequate pregnancy interval and healthy weight of mother especially for those in deprived urban areas.

**Author affiliations**
[1]National Centre for Population Health and Wellbeing Research, Swansea University Medical School, Swansea, UK
[2]Keir Hardie University Health Park, Public Health Wales, Cardiff, UK
[3]Cardiff School of Technologies, Cardiff Metropolitan University, Llandaff Campus, Cardiff, UK
[4]Office for National Statistics, Government Buildings, Cardiff Road, Duffryn, Newport, UK
[5]National Police Chiefs' Council Lead for Mental Health and Age, London, UK
[6]Violence Research Group, School of Dentistry, Cardiff University, Cardiff, UK
[7]Security, Crime, Intelligence Institute, Cardiff University, SPARK, Maindy Road, Cardiff, UK

**Contributors** Planning—conceptualisation: SB, AJ, JE and AB. Data acquisition: The police Public Protection Notification and Maternal Indicator Database data accusation was supported by BR and JE, respectively. The other health data was available in Secure Anonymised Information Linkage, obtained through Information Governance Review Panel (IGRP) request led by SB and AB. Supervision: SB. Conduct—literature review: CT and EM. Methodology: AB and SB. Data preparation: MAR and AB. Formal analysis and investigation: AB. Additional support in analysis: JH and MP. Writing—original draft preparation: AB. Review and editing: CT, MP, JE, EM, HJ, MAR, JH, TLW, BR, SM, AJ and SB. All authors read and approved the final manuscript. SB and AB are responsible for the overall content.

**Funding** This work was funded by Public Health Wales (PHW), grant number (105186). This work was supported by National Institute for Health Research (NIHR), grant number (NIHR133680). This research has been carried out as part of the ADR Wales programme of work. The ADR Wales programme of work is aligned to the priority themes as identified in the Welsh Government's national strategy: Prosperity for All. ADR Wales brings together data science experts at Swansea University Medical School, staff from the Wales Institute of Social and Economic Research, Data and Methods (WISERD) at Cardiff University and specialist teams within the Welsh Government to develop new evidence which supports Prosperity for All by using the Secure Anonymised Information Linkage (SAIL) Databank at Swansea University, to link and analyse anonymised data. ADR Wales is part of the Economic and Social Research Council (part of UK Research and Innovation) funded by ADR UK (grant ES/S007393/1). This work was also supported by the National Centre for Population Health and Well-Being Research (NCPHWR) which is funded by Health and Care Research Wales. This work was supported by Health Data Research UK which receives its funding from HDR UK Ltd (NIWA1) funded by the UK Medical Research Council, Engineering and Physical Sciences Research Council, Economic and Social Research Council, Department of Health and Social Care (England), Chief Scientist Office of the Scottish Government Health and Social Care Directorates, Health and Social Care Research and Development Division (Welsh Government), Public Health Agency (Northern Ireland), British Heart Foundation (BHF) and the Wellcome Trust. This work uses data provided by patients and collected by the National Health Service as part of their care and support. This study used anonymised data held in the SAIL Databank. We would like to acknowledge all the data providers who enable SAIL to make anonymised data available for research. We acknowledge the support provided by South Wales Police. The work conducted does not represent or is it endorsed by the Office for National Statistics.

**Competing interests** None declared.

**Patient and public involvement** Patients and/or the public were not involved in the design, or conduct, or reporting, or dissemination plans of this research.

**Patient consent for publication** Not applicable.

**Ethics approval** Not applicable.

**Provenance and peer review** Not commissioned; externally peer reviewed.

**Data availability statement** Data may be obtained from a third party and are not publicly available. The data have been archived in the Secure Anonymised Information Linkage Databank (https://saildatabank.com/0029)

**ORCID iDs**
Amrita Bandyopadhyay http://orcid.org/0000-0003-2798-4030
Hope Jones http://orcid.org/0000-0003-4312-476X
Emily Marchant http://orcid.org/0000-0002-9701-5991
Simon Moore http://orcid.org/0000-0001-5495-4705
Sinead Brophy http://orcid.org/0000-0001-7417-2858

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
