## [Reviewer comments · BMJ Open]

ARTICLE DETAILS

TITLE (PROVISIONAL)	Weighting of risk factors for low birth weight: A linked routine data cohort study in Wales, UK.
AUTHORS	Bandyopadhyay, Amrita; Jones, Hope; Parker, Michael; Marchant, Emily; Evans, Julie; Todd, Charlotte; Rahman, Muhammad A.; Healy, James; Win, Tint; Rowe, Ben; Moore, Simon; Jones, Angela; Brophy, Sinead

VERSION 1 – REVIEW

REVIEWER	Das , Sarthak All India Institute of Medical Sciences - Deoghar, Pediatrics
REVIEW RETURNED	24-May-2022

GENERAL COMMENTS	statistical analysis can be crosschecked.
---

REVIEWER	Reifsnider, Elizabeth Arizona State University, Phoenix, Arizona, College of Nursing and health Innovation
REVIEW RETURNED	15-Jun-2022

GENERAL COMMENTS	This is an interesting and well-detailed manuscript. My only suggestion is to include more commas and parentheses to separate some ideas and phrases. In several areas I needed to read the sentences several times to understand the meaning. This is one area of note: "Linkage of the cohort with PPN gave a dataset of 5,854 mothers of those who had a PPN call during pregnancy 18% had a LBW child whereas those who did not have PPN call 8.7% had a child with LBW (see Table 1). Mothers with a PPN call during pregnancy had almost 2 times higher risk of having a LBW baby (1.98 (1.39 – 2.81)) than mothers without PPN call after adjusting for confounding factors (see Supplementary Figure 1).
--

REVIEWER	Tata, Laila University of Nottingham, Epidemiology & Public Health
REVIEW RETURNED	27-Sep-2022

GENERAL COMMENTS	-Overall, choice of and reasons for narrow restriction of explanatory variables in this study are unclear and disadvantage the utility of the estimates. More information is needed on the reasoning and (restrictive) choice for the explanatory variables. There is almost no description of the choice/justification of explanatory variables or what the included variables actually are in the methods. The modelled variables are few considering the complexity of the potential causes of birthweight. In particular, there are some key maternal factors missing (e.g., pre-existing and pregnancy-related/initiating hypertension, GI conditions, asthma, CVD). There aren't clear
--

	justifications for use of area in this context either for informing general predictors of LBW. On the other hand, there are some very specific variables, but clarification on the definitions, measurement and ascertainment are needed (e.g., high 'substance use'). -More information is required on the methods of building the MLR as these are not described. Factors such as deprivation compared with diabetes need consideration in modelling. Birth order and multiple birth need particular consideration in terms of their use as 'predictive factors.' Choice of MLR and DT analysis for the main aim need clearer justification if overall aim is to target risk factors considered potentially causal or limitations need to be carefully considered -The abstract results/main conclusion don't consider the DT modelling at all – i.e., analysis is not fully reported/used for drawing conclusions. Reconciling between modelling approaches is needed, particularly the poor performance of the DT which likely relates both to the inclusion/exclusion of explanatory variables and the modelling. -Overall, consideration of the complexity of causal interplays in factors resulting in LBW are not well considered in the design and analysis. The main conclusion paragraph from line 29 on page 12 does not consider this complexity and conclusions drawn may not have practical applications beyond previously known associations. Please note that my expertise is not in machine learning
--	--

VERSION 1 – AUTHOR RESPONSE

Reviewer: 1

Dr. Sarthak Das , All India Institute of Medical Sciences - Deoghar

Comments to the Author:

statistical analysis can be crosschecked.

Response: The authors would like to thank the reviewer for taking time to review the manuscript. The authors would like to highlight that the statistical analysis conducted by the study has been crosschecked by applying two independent statistical methods –

- method 1 – multivariable logistic regression and
- method 2 – decision tree model.

There is significant overlap between the most important risk factors associated with LBW identified by the above two methods. These include maternal smoking status, pregnancy interval, birth order, maternal age, and higher deprivation. Hence having undertaken the necessary cross-checking we can confirm that the results are consistent.

Reviewer: 1

Competing interests of Reviewer: yes

Reviewer: 2

Dr. Elizabeth Reifsnider, Arizona State University, Phoenix, Arizona

Comments to the Author:

This is an interesting and well-detailed manuscript. My only suggestion is to include more commas and parentheses to separate some ideas and phrases. In several areas I needed to read the sentences several times to understand the meaning. This is one area of note: "Linkage of the cohort with PPN gave a dataset of 5,854 mothers of those who had a PPN call during pregnancy 18% had a LBW child whereas those who did not have PPN call 8.7% had a child with LBW (see Table 1). Mothers with a PPN call during pregnancy had almost 2 times higher risk of having a LBW baby (1.98 (1.39 – 2.81)) than mothers without PPN call after adjusting for confounding factors (see Supplementary Figure 1).

Response: The authors would like to thank the reviewer for taking the time to carefully reviewing the paper. We have revised the main manuscript according to the suggestions from the reviewer and have carefully proofread the manuscript to ensure clarity of discussions and eliminate typos.

Reviewer: 2

Competing interests of Reviewer: I have no competing interests

Reviewer: 3

Dr. Laila Tata, University of Nottingham

Comments to the Author:

Response: The authors would like to thank the reviewer for taking time to carefully review the manuscript and providing insightful feedback and detailed comments on the paper. These have helped us to improve our paper and strengthen the discussions around the methodology and discussion. A detailed point-by-point response to the reviewer's comments is included below.

-Overall, choice of and reasons for narrow restriction of explanatory variables in this study are unclear and disadvantage the utility of the estimates. More information is needed on the reasoning and (restrictive) choice for the explanatory variables. There is almost no description of the choice/justification of explanatory variables or what the included variables actually are in the methods. The modelled variables are few considering the complexity of the potential causes of birthweight. In particular, there are some key maternal factors missing (e.g., pre-existing and pregnancy-related/initiating hypertension, GI conditions, asthma, CVD). There aren't clear justifications for use of area in this context either for informing general predictors of LBW. On the other hand, there are some very specific variables, but clarification on the definitions, measurement and ascertainment are needed (e.g., high 'substance use').

Response: We thank the reviewer for this comment and appreciate the opportunity this gives us to explain the rationale behind the explanatory variables of this study. We would like to highlight that the authors conducted a literature review at the commencement of the study which was the foundation of the selection of explanatory variables in this study. At the scoping stage of the project, we identified a recent collaborative study with Public Health Wales (Johnson et al., 2017). The current study was also commissioned by Public Health Wales, but it has gone beyond the risk factors identified in Johnson et al and have considered all additional LBW risk factors reported in literature since 2010. We have developed a report on the literature review, and we have enclosed the exhaustive literature review document as a supplementary file with this submission. Our approach, as a result, has considered a wide range of explanatory and confounding variables which have a plausible causal link to LBW and are potentially modifiable at a population level. For e.g., we have considered maternal physical (such as diabetes, anaemia, intake of Vitamin D and folic acid supplement through prescription) and mental (depression, anti-depressant medication, anxiety, serious mental illness such as bipolar disorder, schizophrenia) health during pregnancy because they were identified in the

literature review as having plausible causal link to LBW.

As a way of addressing the comment and justifying the choice of explanatory variables, we have edited our method section and added necessary details to describe the selection criteria of the explanatory variables in the revised manuscript.

A literature review was conducted at the beginning of the study to identify the explanatory variables associated with LBW. A study by Johnson et al was identified [3] and this provided the framework upon which the current study was developed. The literature review selected

a) any published systematic reviews since 2013 which focused on risk factors identified in Johnson et al.

b) any published systematic reviews since 2010 for all additional risk factors not identified in Johnson et al.

This study therefore considered a wide range of explanatory and confounding variables that have a plausible causal link to LBW and are potentially modifiable at a population level. The literature review to select the explanatory variables has been described in a Supplementary document. In the current study, modifiable risk factors identified from the literature have been derived from routinely collected electronic datasets to build a Welsh e-cohort of the children. The maternal variables related to a childbirth (maternal age, gestational age, child's birth weight and gender, and birth order of the child) were obtained from NCCHD and Maternal Indicator Database (MID). The variables for maternal physical (such as diabetes, anaemia, intake of Vitamin D and folic acid supplement through prescription) and mental (depression, anti-depressant medication, anxiety, serious mental illness such as bipolar disorder, schizophrenia) health during pregnancy were obtained from primary care Welsh Longitudinal General Practice (WLGP) and hospital admissions dataset known as the Patient Episode database in Wales (PEDW). The record of physical assault linked with mothers during pregnancy was obtained from PEDW. The substance misuse database (SMD) provided the information on individuals receiving treatment for alcohol and other substance misuse in Wales. Mothers' who were presenting in this database during pregnancy were considered in the study. Area type (urban/rural) and local authority (LA) under which they lived during the pregnancy and their overall and physical environment quantified in the WIMD were included in this study. A cleaned and harmonised variable of maternal smoking during pregnancy was created based on the data obtained from NCCHD, MIDS and WLGP datasets. The other derived maternal variables include multiple birth flag (to distinguish between singleton and non-singleton), pregnancy interval, and maternal weight. The description of the explanatory variables and their sources have been described in Supplementary Table 1.

A subset of the study population (participants from Rhondda, Cynon, Taf born between June 2016 and 2018) was linked with the Public Protection Notification (PPN) dataset to investigate the impact of the PPN during pregnancy along with other existing risk factors on the risk of LBW. [28]. PPN is an information sharing system, completed by police officers, that compiles incidents of domestic abuse, stalking or harassment. The current study received PPN data from South Wales Police for residents of South Wales local authority Rhondda, Cynon, Taf.

-More information is required on the methods of building the MLR as these are not described. Factors such as deprivation compared with diabetes need consideration in modelling. Birth order and multiple birth need particular consideration in terms of their use as 'predictive factors.' Choice of MLR and DT analysis for the main aim need clearer justification if overall aim is to target risk factors considered potentially causal or limitations need to be carefully considered.

Response: We thank the reviewer for this comment. Further information on the MLR model construction and DT analysis have been included in the manuscript. We would like to highlight that MLR model included birth order of the child and the multiple birth flag. The birth order highlights the sequential birth position of the child for a mother, and it does not vary among the children who were non-singleton (identified from multiple birth flag) in the same family (please see Supplementary Table 1). Hence, they were considered as independent variables in the model and their association with the outcome variable was investigated in the MLR model.

We have edited our Statistical Analysis section to include necessary details as suggested by the reviewer.

The statistical analysis of the current study was carried out using two statistical approaches a) building a holistic regression model to investigate the association between the risk factors and LBW and b) build a predictive model using supervised classification method. Both the methods were capable of handling binary outcome variable. The models that were developed by the above-mentioned methods were built independently, however they both were informed by the same dataset. This enabled us to evaluate and validate the findings of the models and helped to gain insight on the generalisability of the findings.

Logistic regression

A multivariable logistic regression (MLR) model was developed to identify the most important risk factors associated with LBW. The MLR model was built on the overall study population (whole Wales dataset) to examine the associations between all the explanatory and outcome variables. The holistic model considering all the risk factors identified from literature review and selected or derived from routine data includes maternal physical and mental health during pregnancy, maternal smoking, alcohol and other substance misuse record, maternal age, maternal weight, pregnancy interval, living area, LA and deprivation - WIMD score. MLR model also included birth order of the child and the multiple birth flag. The birth order highlights the sequential birth position of the child for a mother, and it does not vary among the children who were non-singleton in the same family (please see Supplementary Table 1), hence, they were considered as independent variables in the model and their association with the outcome variable was investigated in the MLR model. The importance and significance of the risk factors have been evaluated and presented with their adjusted Odds Ratio (aOR) and 95% confidence interval (CI).

The rationale for the choice of MLR and DT for the our statistical analysis to identify the of risk factors for LBW are included in the Discussion section as quoted below.

Discussion:

The study undertook two statistical methods a) regression and b) supervised classification model with the aim that the regression model would identify the risk factors with highest association/Odds Ratio but not frequently observed factors at the population level for e.g., only .09% mothers had diabetes related GP visit during pregnancy, and they had two times higher risk of having a LBW child (2.03 (1.81, 2.28)). However, the DT models consider the number of people affected by the risk factor rather than just strength of association, hence capable of identifying the factors at a population level (such as smoking, deprivation score) that can result in higher risk of LBW.

There are similarities between the findings of our DT models and existing literature utilising machine learning to predict LBW, for e.g., urban living, higher deprivation and poorer families are at higher risk of LBW.

-The abstract results/main conclusion don't consider the DT modelling at all – i.e., analysis is not fully reported/used for drawing conclusions. Reconciling between modelling approaches is needed, particularly the poor performance of the DT which likely relates both to the inclusion/exclusion of explanatory variables and the modelling.

Response: The authors would like to thank the reviewer for this comment. To address the first part of the comment, we have added the following lines regarding the results of DT model in the abstract of the paper.

Abstract:

The MLR model showed that non-singleton children had the highest risk of LBW (adjusted odds ratio 21.74 (95% confidence interval 21.09–22.40)), followed by pregnancy interval less than one year (2.92(2.70–3.15)), maternal physical and mental health conditions including diabetes (2.03(1.81–2.28)), anaemia (1.26(1.16–1.36)), depression (1.58(1.43–1.75)), serious mental illness (1.46(1.04–2.05)), anxiety (1.22, 95% CI 1.08–1.38) and use of anti-depressant medication during pregnancy (1.92(1.20–3.07)). Additional maternal risk factors include smoking (1.80(1.76–1.84)), alcohol-related hospital admission (1.60(1.30–1.97)), substance misuse (1.35(1.29–1.41)) and evidence of domestic abuse (1.98(1.39–2.81)). Living in less deprived area has lower risk of LBW (0.70(0.67–0.72)). The most important risk factors from the DT models include maternal factors (smoking, weight, substance misuse record, age, substance misuse record) along with deprivation - WIMD score, pregnancy interval and birth order of the child.

In regard to 'Reconciling between modelling approaches' and 'poor performance of the DT', we have also included the following paragraphs in the Discussion that compare the findings from the DT models with existing literature and provide a reflection on the performance of the models.

Discussion:

The study undertook two statistical methods a) regression and b) supervised classification model with the aim that the regression model would identify the risk factors with highest association/Odds Ratio but not common or frequently observed factors at the population level for e.g., only .09% mothers had diabetes related GP visit during pregnancy, and they were two times higher risk to have a LBW child (2.03 (1.81 – 2.28)). However, the DT models consider the number of people affected by the risk factor rather than just strength of association, hence capable to identify the common factors (such as smoking, deprivation score).

There are similarities between the findings of our DT models and existing literature utilising machine

learning to predict LBW, for e.g., urban living, higher deprivation and poorer families are at higher risk of LBW [39]. The incidence of LBW in this current work is lower than another research utilising machine learning to predict LBW for e.g., Loreto et al has an incidence of 13.45% in work that builds over 60 different machine learning models [40], Ahmadi et al assess logistic regression and random forests in a cohort with LBW rate of 9.5% [41]. The smaller number of active cases in the dataset the more difficult it is to build a prediction model for, particularly without a set of highly associated input variables. In this study, the singleton DT model correctly predicted 60.41% of all the true positive cases. However, the low positive predictive value of 9.68% indicates that the model assigned a false positive 'LBW' classification for 89.32% cases. This model only includes singleton children and since non-singleton pregnancies are highly associated with LBW, removing this variable from the model has lessened its predictive capability. This is evidenced by the significantly improved positive predictive value (67.36%) for the non-singleton model (table 3). Previous machine learning models appear to show better prediction as they included non-singleton, gestational age (which is in terms of temporal association highly associated with LBW but occurs at the same time as the LBW can be measured) and preeclampsia in third trimester. Also, the differences in the proportion of LBW cases, the variables used, and the cohort sizes in various other studies alter the ability of the model, hence direct comparison of machine learning models across studies can become difficult.

-Overall, consideration of the complexity of causal interplays in factors resulting in LBW are not well considered in the design and analysis. The main conclusion paragraph from line 29 on page 12 does not consider this complexity and conclusions drawn may not have practical applications beyond previously known associations.

Response: The authors would like to thank the reviewer for this comment. The objective of the study is to gain an insight into the modifiable risk factors of LBW and ascertaining their importance and significance as informed by data; the goal of the study was to explore the important explanatory variables affecting LBW rather than establishing a hierarchical causal pathway of factors leading to LBW. Identifying the association through a holistic model is the main objective of this paper. The selection of the explanatory variables is strongly grounded in literature as the study included a set of variables from an exhaustive literature survey that exists in the research domain of LBW. This goes beyond any single literature considering a set of factors but rather takes a holistic view of the analysis by incorporating several experimentally and confounding variables with plausible causal link to LBW (Johnson et al 2017).

In this paper we undertook a comparison of our findings with the reported literature which helped to highlight the similarities and dissimilarities of our findings with a wide range of LBW risk factor. This comparison with reported literature establishes the validity of the factors previously reported for LBW with smaller sets of explanatory variables. The current study considers a superset of the previously reported risk factors and expands the scope and generalisability of those findings. This has also ensured that the LBW risk factors are not considered in isolation and that the risk factors are appropriately adjusted for.

The current study has linked a very wide spectrum of routinely collected nationally representative administrative data to build a database which can facilitate a longitudinal data-linkage study on LBW. This is a very first of its kind in Wales and adds novelty in the research field of LBW and facilitate the addition of vital information into the association between risk factors of LBW through a routine data framework.

Please note that my expertise is not in machine learning

Reviewer: 3

Competing interests of Reviewer: None